# Anti-Scg3 Gene Therapy to Treat Choroidal Neovascularization in Mice

**DOI:** 10.3390/biomedicines11071910

**Published:** 2023-07-06

**Authors:** Chengchi Huang, Liyang Ji, Avinash Kaur, Hong Tian, Prabuddha Waduge, Keith A. Webster, Wei Li

**Affiliations:** 1Department of Ophthalmology, Cullen Eye Institute, Baylor College of Medicine, Houston, TX 77030, USA; 2Everglades Biopharma, LLC, Houston, TX 77098, USA; 3Department of Pharmacology, Vascular Biology Institute, University of Miami Miller School of Medicine, Miami, FL 33136, USA

**Keywords:** age-related macular degeneration, choroidal neovascularization, gene therapy, secretogranin III (Scg3), anti-Scg3 gene therapy, disease-targeted gene therapy, vascular endothelial growth factor (VEGF), anti-VEGF gene therapy

## Abstract

Neovascular age-related macular degeneration (nAMD) with choroidal neovascularization (CNV) is a leading cause of blindness in the elderly in developed countries. The disease is currently treated with anti-angiogenic biologics, including aflibercept, against vascular endothelial growth factor (VEGF) but with limited efficacy, treatment resistance and requirement for frequent intravitreal injections. Although anti-VEGF gene therapy may provide sustained therapy that obviates multiple injections, the efficacy and side effects related to VEGF pathway targeting remain, and alternative strategies to block angiogenesis independently of VEGF are needed. We recently reported that secretogranin III (Scg3) induces only pathological angiogenesis through VEGF-independent pathways, and Scg3-neutralizing antibodies selectively inhibit pathological but not physiological angiogenesis in mouse proliferative retinopathy models. Anti-Scg3 antibodies synergize dose-dependently with VEGF inhibitors in a CNV model. Here, we report that an adeno-associated virus-8 (AAV8) vector expressing anti-Scg3 Fab ameliorated CNV with an efficacy similar to that of AAV-aflibercept in a mouse model. This study is the first to test an anti-angiogenic gene therapy protocol that selectively targets pathological angiogenesis via a VEGF-independent mechanism. The findings support further safety/efficacy studies of anti-Scg3 gene therapy as monotherapy or combined with anti-VEGF to treat nAMD.

## 1. Introduction

Neovascular age-related macular degeneration (nAMD), also known as exudative or wet AMD, is a leading cause of blindness in the elderly populations of developed countries, with the incidence currently estimated to be >1.5 million cases in the US [1]. Wet AMD manifests as choroidal neovascularization (CNV) in the macula and was traditionally treated with thermal laser or photodynamic therapy but with poor efficacy and potential side effects [2]. The advent and approval of anti-angiogenic biological drugs against vascular endothelial growth factor (VEGF), such as ranibizumab and aflibercept, delivered by intravitreal injections represent a major breakthrough for nAMD therapy but with limited efficacy. Clinical trials reported visual acuity improvement (>15 letters) only in 14.6–40.3% of patients treated with anti-VEGF vs. 3.8–5.6% in sham controls [3]. The requirement for frequent intravitreal injections is a major drawback. The half-lives of intravitreally injected ranibizumab and aflibercept are 9 and 11 days, respectively, so monthly injections are required to maintain therapeutic efficacy [4,5]. Frequent intravitreal injections increase the risk of eye complications, including ocular pain, cataract, retinal and vitreous hemorrhage, retinal detachment, endophthalmitis and elevated intraocular pressure [6]. When combined with the physical and psychological stresses associated with repeated ocular injections, the treatment regimens severely impact patient quality of life. Real-world visual outcomes after anti-VEGF therapy often fall short of those in published randomized clinical trials, as a consequence of poor patient compliance, lower injection rates and undertreatment [7,8,9,10].

The possibility for anti-VEGF treatments to be synergized by simultaneously targeting alternative signaling pathways was explored in trials by combining ranibizumab with Fovista or nesvacumab as the antagonist of platelet-derived growth factor (PDGF) or angiopoietin 2 (Ang2), respectively, but failed to achieve endpoints [11,12]. Although safety/efficacy limitations and patient tolerance were cited as reasons for the failures, it seems likely that the common VEGF-dependent signaling pathways of PDGF and Ang2 preclude synergy with anti-VEGF while retaining the limitations of anti-VEGF monotherapy [13,14,15]. Faricimab, a bispecific antibody that simultaneously targets VEGF-A and Ang2, delivered by an intravitreal injection every 12–16 weeks was found to be clinically equivalent to aflibercept for the treatment of nAMD and approved by the FDA on the basis of “no inferiority” [16,17]. The VEGF pathway remains the exclusive target of most ongoing clinical trials [18]. Alternative approaches to replace or synergize with anti-VEGF therapeutics are rarely reported largely because the relevant VEGF-independent regulators of angiogenesis have not been identified.

Gene therapy protocols to deliver aflibercept and ranibizumab are currently in Phase 1/2 clinical trials [19]. The goal is to circumvent the requirement for repeated injections by delivering sustained therapy via a single intraocular injection. The strategy is important but will not address other issues related to limited anti-VEGF efficacy across patient groups. We recently identified secretogranin III (Scg3) as a unique disease-restricted angiogenic factor that drives pathological but not physiological angiogenesis via a VEGF-independent signaling pathway [20]. Scg3-neutralizing antibodies alleviate CNV and dose-dependently synergize with aflibercept in mouse models [21,22,23]. Anti-Scg3 therapeutic antibodies represent potential alternatives or add-ons to anti-VEGF therapy for proliferative ocular diseases but share the same durability obstacles and requirements for repeated injections. The benefits of adapting these therapies to gene therapy are clear and uncontroversial. Here, we investigated the feasibility of anti-Scg3 gene therapy to ameliorate laser-induced CNV in mice and compared efficacies with anti-VEGF gene therapy.

## 2. Materials and Methods

### 2.1. Animals and Materials

C57BL/6J mice (6–8 weeks old, strain # 000664) were purchased from the Jackson Laboratory (Bar Harbor, ME, USA). All animal procedures were approved by the Institutional Animal Care and Use Committee at Baylor College of Medicine (Protocol # AN-8362). An anti-Scg3 clone mT4 monoclonal antibody (mAb) was generated by Everglades Biopharma, LLC (Houston, TX, USA), and it was characterized for Scg3-neutralizing activity, as previously described [20,22]. AAV-CAG-mCherry was purchased from Charles River (Rockville, MD, USA).

### 2.2. Endothelial Cell Proliferation Assay

An endothelial cell proliferation assay was performed to characterize the neutralizing activity of an anti-Scg3 mAb Fab fragment (2.5 μg/mL) in human umbilical vein endothelial cells (HUVECs) in the presence or absence of Scg3 (1 μg/mL), as previously described [23].

### 2.3. Production of Recombinant AAV Vectors

The cDNA coding sequence of anti-Scg3 mouse Fab was amplified using PCR and cloned into an AAV2-CAG plasmid at EcoRI and XhoI sites (Figure 1A). A furin GT2A cleavage site was inserted between the heavy and light chains, and a signal peptide of the human/murine IgG heavy chain was inserted at the N-termini (Figure 1A) [24,25]. The cDNA for aflibercept with a C-terminal FLAG tag and the same N-terminal signal peptide was synthesized by Synbio Technologies (Monmouth Junction, NJ, USA) and cloned into an AAV2-CAG plasmid at EcoRI and XhoI sites (Figure 1A). All plasmids were verified using DNA sequencing. AAV8 was packaged and purified by SignaGen (Frederick, MD, USA) using CsCl gradient centrifugation, followed by dialysis against phosphate-buffered saline (PBS). The viral genome (vg) was titrated via qPCR using a standard curve and subsequently verified using digital PCR (dPCR) (Applied Biosystems/Thermo Fisher Absolute Q, Waltham, MA, USA) [26].

### 2.4. Characterization of AAV Plasmids and Viral Vectors

Human embryonic kidney 293 cells (HEK293) were seeded on 6-well plates at 2 × 10^5^ cells/well in Dulbecco’s modified Eagle’s minimum essential medium (DMEM) (Gibco/Thermo Fisher Scientific, Waltham, MA, USA) supplemented with 10% fetal bovine serum (FBS) (Gibco), 1x GlutaMAX (Thermo Fisher) and 1% penicillin/streptomycin (Gibco) and incubated at 37 °C overnight in a humidified atmosphere with 5% CO_2_. The cells were transduced with AAV-anti-Scg3Fab, AAV-aflibercept or AAV-mCherry at 5 × 10^6^ vg/mL. The medium was replaced with a serum-free 293SFM II medium (Thermo Fisher) the following day. Five days post-transduction, the conditioned medium was collected and concentrated using an Amicon^®^ Ultra-4 Centrifugal Filter Unit (UFC801008, Millipore Sigma, St. Louis, MO, USA). An enzyme-linked immunosorbent assay (ELISA) was performed with pre-immobilized Scg3 (5 μg/mL, 100 μL/well, Sino Biological, Wayne, PA, USA), recombinant human VEGF (VEGF, 5 μg/mL, R&D Systems, Minneapolis, MN, USA) or bovine serum albumin (BSA, Sigma). Bound anti-Scg3 mAb and aflibercept were detected with biotin-conjugated anti-FLAG M2 mAb and horseradish peroxidase (HRP)-conjugated streptavidin (Sigma), followed by a colorimetric assay [27].

### 2.5. AAV Administration

Mice were anesthetized via an intraperitoneal (i.p.) injection of ketamine (40 mg/kg body weight, Covetrus North America, Portland, ME) and xylazine (8 mg/kg, Akom, Lake Forest, IL, USA). AAV-anti-Scg3Fab, AAV-aflibercept or AAV-mCherry (5.0 × 10^8^ vg/1 μL/eye) was blind-coded and intravitreally injected.

### 2.6. Laser-Induced Choroidal Neovascularization

Mice were subjected to laser photocoagulation to induce CNV at 1 or 4 months after the AAV injection by using the following procedures: For pupil dilation, anesthetized mice received a topical eye drop of 1% tropicamide (Akorn, Lake Forest, IL, USA) and 2.5% phenylephrine (Paragon BiTeck, Portland, ME, USA). A green laser beam (532 nm, 240 mW, 150 ms, 50 µm spot) was applied to the retina around the optic disk (4 spots/retina) using a Micron IV retinal imaging system (Phoenix Research Labs, Pleasanton, CA, USA). Gaseous bubbles formed at laser spots indicated the rupture of Bruch’s membrane. Lesions with retinal hemorrhage on Day 0 and linear or fused lesions on Day 7 were excluded.

### 2.7. Fluorescein Angiography

Fluorescein angiography was conducted on Day 7 after laser photocoagulation. All fluorescein angiography images were taken 6 min after an injection of fluorescein sodium (0.1 mL/mouse, 2.5%, Akorn) in anesthetized mice with standardized instrument settings using a Spectralis Tracking OCTA system (Heidelberg Engineering, Franklin, MA, USA). Fluorescein angiography images were analyzed using ImageJ software (NIH). The area and intensity of the laser spots were normalized to cognate the entire viewing field of the eye. After fluorescein angiography, the retinal pigment epithelium (RPE)–choroid–sclera eyecups (RPE eyecups) were isolated from the euthanized mice, fixed, stained with Alexa Fluor 488-isolectin B4 (AF488-IB4, 10 μg/mL, Thermo Fisher), flat-mounted, and analyzed using a Keyence BZ-X810 structured illumination microscope (SIM) and Keyence software.

### 2.8. Immunohistochemistry

The mice with CNV were euthanized with CO_2_ inhalation after fluorescein angiography, and they were immediately perfused intracardially with PBS, followed by 4% paraformaldehyde (PFA) and eye enucleation. The anterior segments, including the cornea and lens, were removed to yield RPE eyecups that were embedded in the optimal cutting temperature (OCT) compound (Tissue-Tek; Miles Scientific, Napierville, IL, USA) and cryosectioned with a 10 μm thickness. The retinal sections were immunostained with anti-FLAG mouse M2 mAb (Sigma, #F1804, dilution 1:200), followed by Alexa Fluor 594-conjugated anti-mouse IgG F(ab’)_2_ (Cell Signaling, Danvers, MA, USA; #8889S; dilution 1:1000) and Hoechst staining, and analyzed using SIM microscopy.

### 2.9. Western Blot

Western blots were performed as previously described [20]. Briefly, the total protein was isolated from the retinas and homogenized in a RIPA buffer (Sigma) supplemented with a protease inhibitor cocktail (Sigma, Cat. #P8340). The total protein was quantified using a BCA protein assay Kit (Thermo Fisher), separated by SDS-PAGE (20 µg/lane), and transferred onto nitrocellulose membranes (Millipore). The membranes were probed with anti-FLAG M2 mAb (1:1000) and a horseradish peroxidase (HRP)-conjugated secondary antibody (Ab) for chemiluminescence signal detection, followed by stripping and reprobing with anti-β-actin mAb.

### 2.10. Statistical Analysis

Data are expressed as mean ± SEM. A statistical analysis was performed using a one-way ANOVA test. *p* < 0.05 was considered significant.

## 3. Results

### 3.1. Construction and In Vitro Characterization of AAVs

We chose anti-Scg3 mAb over the related humanized antibody (hAb) [20,23] for this project to minimize potential mouse anti-human IgG Ab that may attenuate the efficacy of anti-Scg3 gene therapy. We constructed a mouse anti-Scg3 mAb Fab fragment in an AAV2 vector with a CAG promoter [28]. The heavy and light chains contained the same human/murine IgG heavy chain signal peptide, separated by a furin GT2A cleavage site (Figure 1A) [24,25]. A FLAG tag was attached to the C-terminus of the light chain. AAV-aflibercept was constructed in a similar fashion. Anti-Scg3 mAb Fab was verified for its neutralizing activity to block Scg3-induced endothelial proliferation of HUVECs in a culture before AAV construction (Figure 1B). After packaging into AAV8, purified AAVs were used to transduce HEK293 cells. Conditioned media were collected and analyzed for binding activity to immobilized VEGF or Scg3 using ELISA. The results show that the HEK293 cells transduced with AAV-anti-Scg3Fab and AAV-aflibercept secreted functionally active anti-Scg3 Fab and aflibercept with corresponding binding activity to Scg3 and VEGF, respectively (Figure 1C). The results suggest that the heavy and light chains of anti-Scg3 Fab are appropriately processed by the furin protease cleavage and assembled into a functionally active Fab fragment. By contrast, a similar construct of anti-Scg3 Fab without the second signal peptide produced non-functional Fab without Scg3-binding activity.

### 3.2. Transgene Expression In Vivo

To characterize gene expression in mouse retinas, we intravitreally injected AAV-anti-Scg3Fab, AAV-aflibercept and AAV-mCherry into mice and analyzed gene expression one month post-injection. Western blot detected the expressions of the FLAG-tagged anti-Scg3 Fab light chain and aflibercept at approximately 27 kDa and 67 kDa, respectively, under reduced and denaturing conditions (Figure 1D). The predicted molecular weight was 26.7 kDa for anti-Scg3 light chain-FLAG and 51.4 kDa for aflibercept-FLAG without glycosylation. The size of the anti-Scg3 light chain also confirmed the cleavage by the furin protease. According to FDA drug information, the aflibercept dimer produced from CHO cells is 97 kDa (monomer 48.5 kDa) but migrates as 115 kDa, with the additional 15% of the total molecular weight attributed to glycosylation [29]. Therefore, aflibercept expressed in the mouse retina by AAV-aflibercept was glycosylated with a ~30% increase in the molecular weight.

To independently validate the transgene expression pattern in the mouse retinas, we performed immunohistochemistry one month after the AAV injection using anti-FLAG mAb. The assay detected the expressions of FLAG-tagged anti-Scg3 Fab and aflibercept in the retinal ganglion cell (RGC) layer, inner plexiform layer (IPL), inner nuclear layer (INL), outer plexiform layer (OPL), photoreceptor inner segments (PISs) and RPE. A reduced expression was also detected in the outer nuclear layer (ONL) and photoreceptor outer segments (POSs) (Figure 2). These expression patterns suggest that the transgenes are expressed throughout the entire retina. No or a minimal FLAG signal was detected for AAV-mCherry or retinal sections without the primary Ab, supporting the signal specificity.

### 3.3. Anti-Angiogenic Gene Therapy to Inhibit CNV

To compare efficacies, we intravitreally injected AAV-anti-Scg3Fab, AAV-aflibercept and AAV-mCherry into mice, followed by CNV induction one month after the AAV injection. We quantified the CNV leakage and related therapeutic efficacy in anesthetized mice using fluorescein angiography 7 days after CNV induction. The results indicate that AAV-anti-Scg3Fab significantly ameliorated CNV leakage in terms of the leakage area and intensity (Figure 3A–C). AAV-aflibercept as a positive control reduced the CNV leakage area and intensity by the same degree as the vehicle control vector AAV-mCherry.

To quantify CNV lesions using histopathology, we euthanized the mice after fluorescein angiography, and we isolated and stained the RPE–choroid–sclera eyecups with AF488-IB4 to label CNV vessels. An SIM microscopy analysis confirmed that AAV-anti-Scg3Fab and AAV-aflibercept significantly reduced CNV lesion size and 3D volume with similar efficacy (Figure 3D–F).

### 3.4. Long-Term Efficacy in Alleviating CNV

To investigate whether gene therapy in this model provides long-term therapeutic benefits, we induced CNV in mice 4 months after AAV transduction and analyzed therapeutic efficacy via fluorescein angiography and immunostaining with AF488-IB4 7 days after the CNV induction. The results show that both AAV-anti-Scg3Fab and AAV-aflibercept significantly and similarly reduced CNV lesion size and 3D volume (Figure 4D–F). However, unlike the 1-month protocol, the CNV leakage size and intensity were not significantly reduced by either treatment at 4 months (Figure 4A–C). Compared to transgene expression at one month after the AAV injection, immunohistochemistry revealed that the transgene expression at 4 months was markedly reduced (Figure 2 vs. Figure 5).

## 4. Discussion

We provide the first evidence that an Scg3 antagonist with a signaling pathway distinct from that of VEGF reduced CNV lesion size and leakage in a gene therapy protocol that is quantitatively equivalent to AAV-aflibercept administered according to the same regimen and gene dosing. Both genes were delivered by AAV8, a serotype appropriate for ocular indications [25]. The vectors delivered equivalent levels of expression at one month after intravitreal injections in mice with 5.0 × 10^8^ vg/eye and generated secreted gene products that displayed the expected selective binding to immobilized VEGF and Scg3 ligands. Immunohistochemistry revealed common distributions of the respective AAV-aflibercept and AAV-anti-Scg3Fab gene products in all retinal layers (Figure 2). AAV-anti-Scg3Fab and AAV-aflibercept reduced the CNV 3D volume and maximal lesion area by the same degrees, indicating equivalent efficacy at both short and extended time intervals after AAV transduction. However, protection against CNV leakage was significantly effective only in the 1-month group and not at 4 months after transduction. We attribute the diminished efficacy to the relatively low vector dose, resulting in a gradual reduction in transgene expression over time (Figure 2 vs. Figure 5). The reduced transgene expression over time may be due to epigenetic regulations after AAV integration. Although the sustained expression of AAV in ocular tissues is well established [25,30,31,32], Liu et al. reported a strict dose-dependent expression and therapeutic efficacy of AAV8 anti-VEGF Fab at 1 month in a mouse CNV model with a markedly reduced efficacy at doses below ~10^9^ vg/eye [25]. This is also consistent with the high dose of 2 × 10^12^ vg/eye used to express AAV-aflibercept in African green monkeys, equivalent to ~2 × 10^9^ vg/eye in mice assuming a factor of ~660× for primate versus mouse vitreous volume [30,31,32,33]. Such a low-dose instability may account for the variable results seen in some gene therapy trials. In a Phase I trial of intravitreal AAV2-sFLT01, gene expressions in the aqueous humor of patients injected with <2 × 10^10^ vg/eye were below the limits of detection [34]. An insufficient transgene expression of 1 × 10^11^ vg/eye AAV2 via subretinal delivery in patients is suspected for the low efficacy and failure of sFLT01 clinical trials [35,36]. The results suggest a narrow window or threshold of the AAV vector dose to achieve sustained therapeutic gene expression [25]. In future studies, we will characterize the dose–response curves of AAV-anti-Scg3Fab and AAV-aflibercept in parallel to determine the dose requirement for persistent anti-Scg3 gene therapy. Additionally, we will investigate whether AAV2 and AAV2.7m8 further improve therapeutic duration [34,37].

Aflibercept contains the Ig domains of VEGFR1 and VEGFR2 fused to the Fc region of human IgG and functions as a soluble VEGF decoy receptor that binds and neutralizes VEGF-A, VEGF-B and placental growth factor (PIGF) [38]. Aflibercept is currently a first-line therapy for nAMD, macular edema following retinal vein occlusion, DR and diabetic macular edema [38]. The related and competing first-line VEGF-neutralizing reagents that are also FDA-approved for the same or similar ocular indications include ranibizumab; bevacizumab (used off-label for nAMD); brolucizumab; and more recently faricimab, a bispecific anti-VEGF and anti-Ang2 Ab [39]. To achieve stable efficacy for nAMD, all the approved drugs of this category require frequent intraocular injections, a requirement that adversely affects patient quality of life and treatment compliance [9,10]. Multiple studies have confirmed the requirement for rigid adherence to injection regimens and the loss or reversal of therapy caused by non-compliance [7,8]. Consequently, intense efforts are underway to circumvent monthly injections by devising sustained-release technologies and/or gene therapy that promises persistent therapy via a single intraocular injection [37]. After the failure of early efforts to translate endogenous angiogenesis inhibitors, such as angiostatin/endostatin and pigment epithelial-derived factor, into gene therapy applications, the field focused on adapting the current FDA-approved repertoire of VEGF pathway blockers [40]. After promising preclinical results [37], ADVM-022, an optimized AAV2 vector encoding aflibercept, is currently in a Phase 2 trial to treat nAMD [19,41]. In a second ongoing Phase I/IIa trial, RGX-314, an AAV8 vector expressing a mAb fragment similar to ranibizumab, delivered by a subretinal injection is being evaluated in patients with nAMD [36]. The safety/efficacy results at 1.5 years reported an improved or stabilized best-corrected visual acuity (BCVA), a reduced central retinal thickness (CRT) and markedly decreased requirements for supplemental anti-VEGF injections [42].

The progression of anti-angiogenesis protocols to gene therapy clinical trials promises to relieve the constraints of frequent intravitreal injections for patients with nAMD but fails to address the confounding issues of suboptimal therapy and other adverse side effects associated with current anti-VEGF therapy. Recent studies reported that VEGF inhibitors are effective in alleviating CNV in young but not aged animal models [43,44], implying that the promised efficacy of anti-VEGF in young animals may not be translated quantitatively to aged patients with nAMD. The alternating use and/or combination of these reagents with inhibitors of VEGF-dependent accessory factors, such as PDGF, Ang2 and semaphorin 6A [45], are unlikely to significantly improve efficacy as next-generation pharmacology or gene therapy for CNV indications [11,12]. Central to this dilemma is that the independence of VEGF signaling is a likely prerequisite for synergistic combinations with current anti-VEGF protocols, and, except for anti-Scg3, no inhibitors of pathological angiogenesis that work independently of VEGF signaling have been described.

Scg3 was discovered by our group using a novel comparative ligandomics technology to screen for disease-restricted ligands in mouse models of DR and CNV [20,22]. Scg3 is a disease-restricted angiogenic factor that selectively binds to diseased but not healthy vessels. Scg3-neutralizing Abs alleviated CNV, DR and retinopathy of prematurity (ROP) in mouse models with an efficacy equivalent to that of aflibercept [20,21,22,23,46,47,48]. The inhibition of angiogenesis by anti-Scg3 Abs in all cases was independent of VEGF, consistent with separate angiogenic signaling pathways and synergy between anti-Scg3 hAb and aflibercept [23]. In Scg3, we appear to have uncovered a hitherto invisible but long-sought-after disease-restricted proangiogenic pathway that operates in parallel but independent of VEGF in pathological states [20]. Such a property that restricts Scg3 actions to pathological angiogenesis is consistent with our findings of safety and wide therapeutic windows of anti-Scg3 vs. anti-VEGF [46,47]. Because, like anti-VEGF biologics, intraocular anti-Scg3 hAb is expected to have a short therapeutic duration, and clinical applications will also require repeated injections. Therefore, translation from protein pharmacology to gene therapy is an important next step to relieve these constraints.

Safety concerns have overshadowed the progression of anti-VEGF gene therapy since its inception. In addition to indiscriminate binding to diseased and healthy vessels, VEGF possesses neurotrophic and neuroprotective properties that promote neuronal growth and survival [49,50,51]. We and others have shown that intravitreal aflibercept induces abnormalities in electroretinography (ERG) and retinal structure in animal models [47,48,52]. Similar adverse effects on retinal function and structure were also reported in patients treated with VEGF inhibitors in some clinical studies [53,54,55,56]. Clinical studies also revealed that intravitreal anti-VEGF agents can interfere with the function of the central nervous system [57,58,59]. Long-term anti-VEGF therapy for nAMD may increase the risk of geographic atrophy [60]. A recent study reported trends of b-wave suppression by AAV-aflibercept in adult non-human primates 19 months after gene therapy [37]. Aged neurons in patients with AMD may be more susceptible to low-ambience VEGF ligands caused by anti-VEGF therapy, thereby contributing to a limited improvement in visual acuity [3]. Indeed, the possible neurotoxicity of anti-VEGF was implied in clinical trials of nAMD and diabetic macular edema (DME), in which high-dose ranibizumab inversely reduced long-term visual acuity despite improving vascular symptoms [61,62]. It is unclear whether neurotoxicity is caused by the direct effects of VEGF blockade on neurons or indirectly through the suppression of healthy vessels. It seems possible that patients with nAMD are more sensitive to manipulations of VEGF homeostasis due to age and comorbidities. Safety concerns related to chronic VEGF suppression prompted the testing of multiple strategies to regulate anti-VEGF gene therapy [63,64,65]. Because our studies indicate that Scg3 regulates only pathological angiogenesis, with no effect on healthy vessels or neurons [24,26], such safety issues may not apply, and long-term constitutive gene therapy with anti-Scg3 hAb is expected to be safe. Our findings warrant further investigation to compare the efficacy and safety of optimized AAV-anti-Scg3 and AAV-anti-VEGF for monotherapy or combination therapy in large animal models.

## Figures and Tables

**Figure 1 biomedicines-11-01910-f001:**
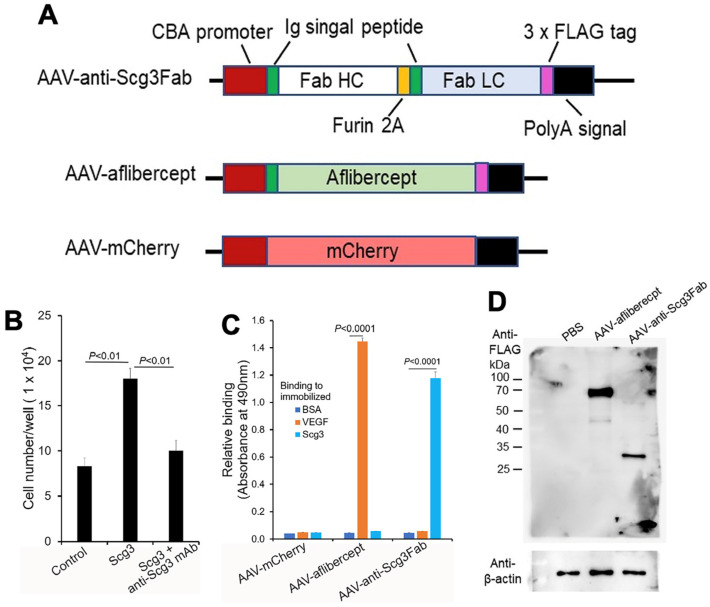
Construction and characterization of AAVs. (**A**) Design of AAV-anti-Scg3Fab, AAV-aflibercept and AAV-mCherry. (**B**) Neutralizing activity of anti-Scg3 mAb to block Scg3-induced proliferation of HUVECs. *n* = 3 wells/group. (**C**) Functional validation of AAV-mediated transgene expression. HEK293 cells were transduced by indicated AAVs. Conditioned media were analyzed for binding activity to immobilized Scg3, VEGF or BSA (negative control) using ELISA. *n* = 3 wells/group. (**D**) AAV-mediated FLAG-tagged transgene gene expression in mouse retinas. Indicated AAVs were intravitreally injected into mice, and retinas were isolated one month after AAV injection and analyzed using Western blot with anti-FLAG mAb. ± SEM, one-way ANOVA test.

**Figure 2 biomedicines-11-01910-f002:**
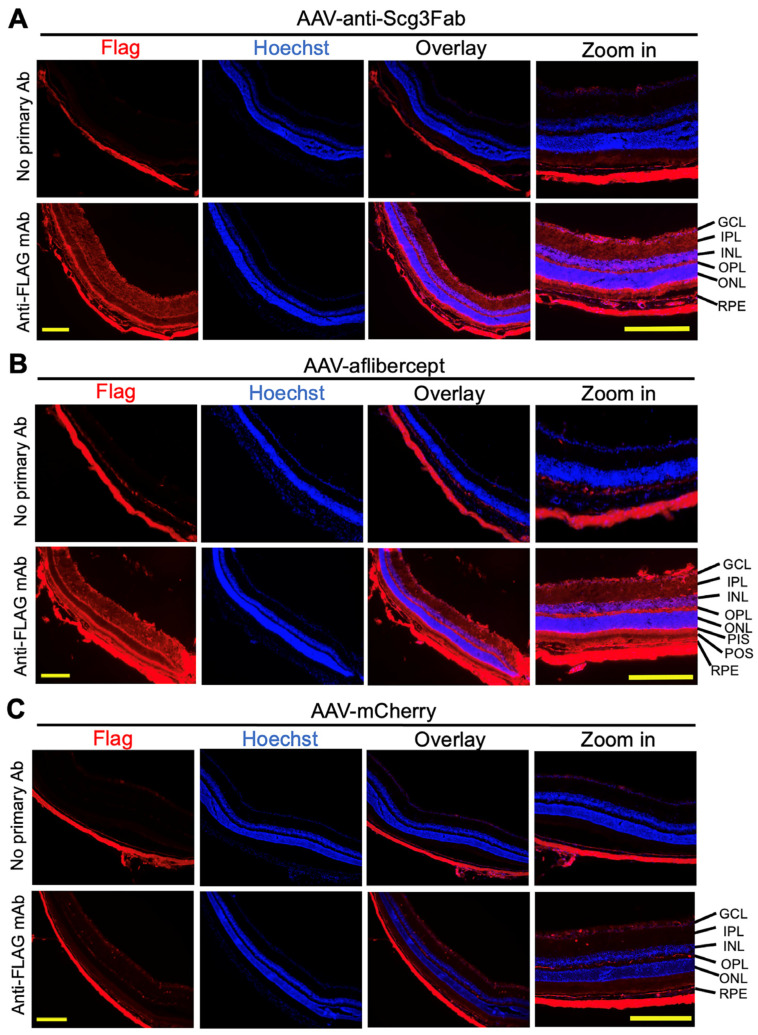
Immunohistochemistry to detect AAV-mediated gene expression in mouse retinas. AAV-anti-Scg3Fab (**A**), AAV-aflibercept (**B**) and AAV-mCherry (**C**) were intravitreally injected into mice, and eyes were isolated from euthanized mice 1 month after AAV injection. Immunohistochemistry was performed using anti-FLAG mAb. Yellow scale bar = 200 µm. GCL, retinal ganglion cell layer; IPL, inner plexiform layer; INL, inner nuclear layer; OPL, outer plexiform layer; ONL, outer nuclear layer; PIS, photoreceptor inner segment; POS, photoreceptor outer segment; RPE, retinal pigment epithelium.

**Figure 3 biomedicines-11-01910-f003:**
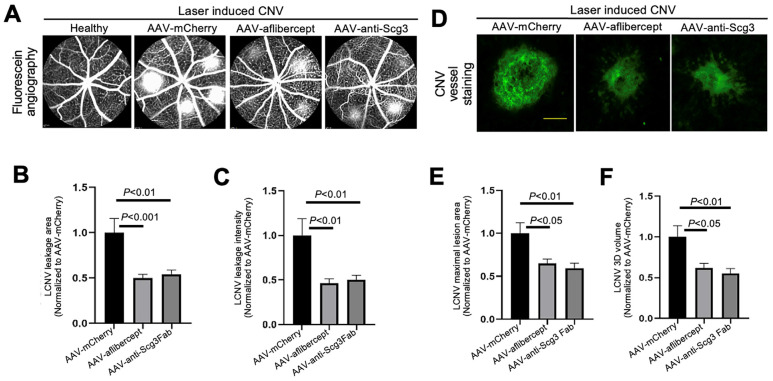
Short-term therapeutic efficacy of AAV-anti-Scg3 and AAV-aflibercept in alleviating CNV. Indicated AAVs (blind-coded) were injected intravitreally into mice. After one month, mice were treated with laser photocoagulation to induce CNV. (**A**) Representative images of fluorescein angiography performed in anesthetized mice 7 days post-laser. (**B**) Quantification of CNV leakage area in (**A**). (**C**) Quantification of CNV leakage intensity in (**A**). *n* = 27 laser spots in 9 eyes (AAV-mCherry), 28 laser spots/8 eyes (AAV-aflibercept) and 32/10 (AAV-anti-Scg3Fab). (**D**) Representative images of eyecups isolated from mice 7 days post-laser and immunostained with Alexa Fluor 488-isolection B4 (AF488-IB4). (**E**) Quantification of CNV maximal lesion area in (**D**). (**F**) Quantification of CNV 3D volume in (**D**). *n* = 29 laser spots/9 eyes (AAV-mCherry), 21/8 (AAV-aflibercept) and 25/10 (AAV-anti-Scg3Fab). ± SEM; one-way ANOVA test. Scale bar = 100 µm.

**Figure 4 biomedicines-11-01910-f004:**
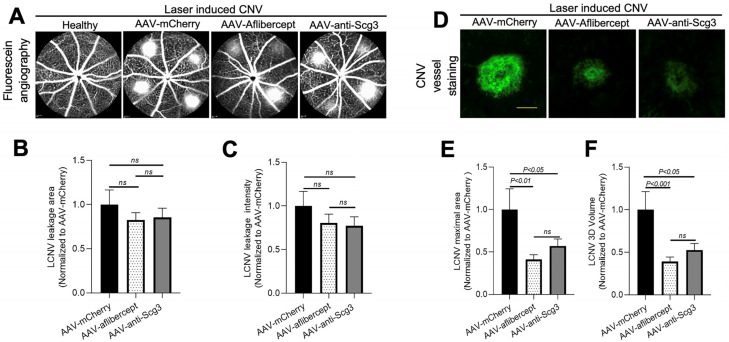
Long-term therapeutic efficacy of AAV-anti-Scg3 and AAV-aflibercept in alleviating CNV. Indicated AAVs (blind-coded) were injected intravitreally into mice. After 4 months, mice were treated with laser photocoagulation to induce CNV and analyzed, as described in Figure 3. (**A**) Representative images of fluorescein angiography. (**B**) Quantification of CNV leakage area in (**A**). (**C**) Quantification of CNV leakage intensity in (**A**). *n* = 19 laser spots in 6 eyes (AAV-mCherry), 30 laser spots/9 eyes (AAV-aflibercept) and 26/8 (AAV-anti-Scg3Fab). (**D**) Representative images of eyecups isolated from mice 7 days post-laser and immunostained with AF488-IB4. (**E**) Quantification of CNV maximal lesion area in (**D**). (**F**) Quantification of CNV 3D volume in (**D**). *n* = 14/6 (AAV-mCherry), 25/9 (AAV-aflibercept) and 24/8 (AAV-anti-Scg3Fab). ± SEM; one-way ANOVA test. Scale bar = 100 µm.

**Figure 5 biomedicines-11-01910-f005:**
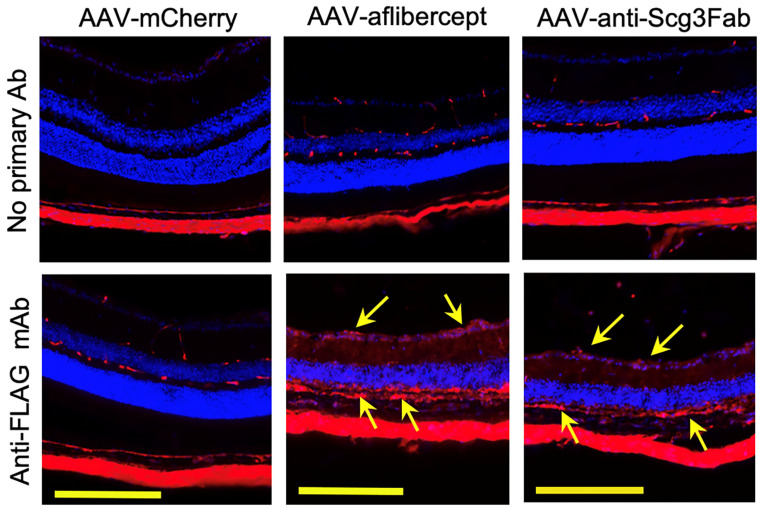
Immunohistochemistry to detect long-term AAV-mediated gene expression in mouse retinas. AAV-anti-Scg3Fab, AAV-aflibercept and AAV-mCherry were intravitreally injected into mice, and eyes were isolated from euthanized mice 4 months after AAV injection. Immunohistochemistry was performed using anti-FLAG mAb. Yellow arrows indicate FLAG^+^ transgene signals. Yellow scale bar = 200 µm.

## Data Availability

All data related to this study are reported herein.

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
