# Peer review of "Anti-Scg3 Gene Therapy to Treat Choroidal Neovascularization in Mice"

_biomedicines, 2023, doi:10.3390/biomedicines11071910_

Round 1
Reviewer 1 Report
It is a very impressive study. A new anti-Scg3 Fab was developed to alleviate CNV. But here are some questions that need to be concerned.
1. In previous study of the authors (Ref.20, Secretogranin III as a disease-associated ligand for antiangiogenic therapy of diabetic retinopathy), Scg3-neutralizing antibody was proved to alleviate retinal vascular leakage in diabetic mice with high efficacy. In prevent study, similar efficiency with AAV-aflibercept was achieved by AAV-anti-Scg3. Similar work were done to prove the efficacy of anti-VEGF. But the superiority of anti-Scg3 Fab is not demonstrated in this study.
2. In this study, the efficacy of AAV-anti-Scg3 decreased after 4 months post AAV transduction. Does it mean repeat injections are required for AAV-anti-Scg3? If so, there is no big difference with common anti-VEGF drugs.
Author Response
Response to Reviewer #1
- But the superiority of anti-Scg3 Fab is not demonstrated in this study.
Response: Our comparative ligandomics identified Scg3 as a disease-restricted angiogenic factor that selectively binds to diseased but not healthy vessels, whereas VEGF binds equally to both diseased and healthy vessels (PMID: 28330905). The superiority of anti-Scg3 is still under investigation and is yet to be fully appreciated for the novel disease-targeted therapy. Thus far, we revealed the following unique properties of anti-Scg3 Fab vs. aflibercept.
- Anti-Scg3 selective inhibits pathological but not physiological angiogenesis, whereas aflibercept indiscriminately suppresses angiogenesis (PMID: 35006382).
- Combination of anti-Scg3 and anti-VEGF synergistically inhibits CNV (PMID: 34918375). To our knowledge, few studies, including Phase 3 trials of faricimab (PMID: 35085502), have demonstrated even additive efficacy improvement by co-inhibition of VEGF and other angiogenic factors (PMID: 28195611).
- Based on their distinct binding activity patterns to diseased vs. healthy vessels, we predicted and identified an VEGF-independent receptor as the major and likely sole Scg3 receptor (Scg3R) (unpublished data), thereby supporting synergistic combination of anti-VEGF and anti-Scg3. In contrast, many other conventional angiogenic factors, such as PDGF and angiopoietin 2, may share similar signaling pathways with VEGF (PMID: 10197615)
- Recent studies reported that anti-VEGF is ineffective to alleviate CNV in aged animals (PMID: 32678293; 25973441). We confirmed that aflibercept is effective to ameliorate laser-induced CNV (LCNV) in young but not aged mice (data not shown). This may be partially responsible for limited efficacy of anti-VEGF to alleviate CNV and improve visual acuity in patients with wet age-related macular degeneration (AMD). However, anti-Scg3 ameliorates LCNV in both young and aged mice with similar high efficacies (unpublished data). This brings a hope that anti-Scg3 may improve efficacy to treat wet AMD in aged patients.
- Many AMD patients with poor response to anti-VEGF often develop subretinal fibrosis (SRF) that may limit improvement in visual acuity (PMID: 35468694). Indeed, we confirmed that aflibercept is completely ineffective to inhibit SRF in aged mice with LCNV, whereas anti-Scg3 efficiently resolves both LCNV and related SRF (unpublished data).
- In a mouse model of alkali-burned cornea, anti-VEGF inhibits corneal neovascularization (CoNV) but not resolve corneal opacity, where anti-Scg3 significantly ameliorates both CoNV and opacity (unpublished data).
The published superiority of anti-Scg3 in a and b has been discussed in the manuscript. All other unique features of anti-Scg3 are unpublished data and cannot be reported at this time. Additionally, these unique features of anti-Scg3 are out of the scope of the manuscript that is to investigate the feasibility of anti-Scg3 gene therapy.
- In this study, the efficacy of AAV-anti-Scg3 decreased after 4 months post AAV transduction. Does it mean repeat injections are required for AAV-anti-Scg3? If so, there is no big difference with common anti-VEGF drugs.
Response: Since current therapies (except Vabysmo) necessitate repeated injections every 4-6 weeks, a reduction of injection frequency to every 16 weeks would be a significant improvement to current therapies. Additionally, the approach addresses patients that do not respond to anti-VEGF therapies.
This is our first study to investigate the feasibility of anti-Scg3 gene therapy with AAV-aflibercept as a positive control. Previous studies reported that single injection of AAV-aflibercept (ADVM-022; AAV2.7m8) in non-human primates resulted in sustained expression up to 30 months (PMID: 33532145). Therefore, we will investigate various approaches, including AAV2, AAV2.7m8 and high AAV titers, to extend the therapeutic duration of AAV-anti-Scg3Fab and AAV-aflibercept in the next study (see our response to Reviewer #2 Comment #4).

Reviewer 2 Report
Huang et al are reporting the effects of anti-Scg3 gene therapy in a preclinical model of neovascular AMD . The authors demonstrate non-inferiority with gene therapy delivered aflibercept when delivered prophylactically in LCNV mice. The authors argue this work could provide proof of concept evidence to explore this topic further as a potential future therapy for nAMD patients.
Major Concerns:
1) I have serious reservations about this biofactory approach using intravitreal delivered AAV8; AAV8 is not optimized for IVT delivery compared with other serotypes (AAV2 variants). The authors show images with diffuse staining of Scg3 and aflibercept 1 and 4 months post injection, however Scg3 and aflibercept are secreted proteins so the PKPD is impossible to understand. ISH would be a better approach to show the extent of transduction and protein production. The mCherry control could be useful here, but staining appears weak or non-existent. Higher magnification images might be helpful. The explanation in the discussion about why protein production is reduced after 4 months (lines ~275-287) is not compelling. The issue could be poor transduction, tox-related issues, or titer and each should be examined and tested empirically.
2) As with any gene therapy approach, tox analyses are paramount. Did the authors seen any signs of inflammation or toxicity? This topic is as important as efficacy and should be carefully analyzed.
3) What is the status of injected anti-Scg3 therapies? The field will need to see much more efficacy/safety data from independent labs and in large animal models, probably even in a clinical trial, with a therapeutic protein before considering gene therapy. Any discussion today about anti-Scg3 gene therapies is, in my opinion, premature.
4) The statements regarding the limitations of anti-VEGFs in the introduction and discussion are overly pessimistic compared to reality. Furthermore, it is confusing that the authors suggest an anti-Scg3 + anti-VEGF combination therapy considering the list of concerns about anti-VEGF gene therapies listed in the last paragraph of the discussion.
Author Response
Response to Reviewer #2
- I have serious reservations about this biofactory approach using intravitreal delivered AAV8; AAV8 is not optimized for IVT delivery compared with other serotypes (AAV2 variants).
Response: As the reviewer mentioned correctly, this was a first proof-of-concept or feasibility study. We will investigate AAV2, AAV2.7m8 and high titer in next study (see our response to Reviewer #1 Comment #2).
- The authors show images with diffuse staining of Scg3 and aflibercept 1 and 4 months post injection, however Scg3 and aflibercept are secreted proteins so the PKPD is impossible to understand. ISH would be a better approach to show the extent of transduction and protein production.
Response: We agree that immunostaining is not suitable to determine pharmacokinetics (PK). PK will be further analyzed by ELISA in future studies. The purpose of immunohistochemistry (IHC) to detect total transgene expression in the retina is to assess underlying causes responsible for the partially reduced efficacy after 4 months. ISH (in situ hybridization) is to detect transgene expression at the mRNA but not protein level. mRNA expression may not always correlate with protein expression (PMID: 36343915), and mRNA is also susceptible to degradation during ISH. Therefore, we chose to detect transgene protein expression by IHC to assess total transgene protein expression in the retina.
- The mCherry control could be useful here, but staining appears weak or non-existent.
Response: mCherry is a negative control that is not tagged with FLAG. Therefore, anti-FLAG detection of mCherry is supposed to show weak and non-existent signals.
- The explanation in the discussion about why protein production is reduced after 4 months (lines ~275-287) is not compelling. The issue could be poor transduction, tox-related issues, or titer and each should be examined and tested empirically.
Response: We agree with the reviewer that AAV8, titer and other issue should be examined. The titer issue has been extensively discussed in Discussion. We do not believe that transgene toxicity is an issue because both aflibercept and Scg3 have the similar partial reduction of efficacy after 4 months. Therefore, we revised Discussion by adding the following sentence. “Additionally, we will investigate whether AAV2 and AAV2.7m8 to further improve therapeutic duration [34,39].” (see our response to Reviewer 1# Comment 2 and Reviewer #2 Comment #1 and 5).
- As with any gene therapy approach, tox analyses are paramount. Did the authors see any signs of inflammation or toxicity? This topic is as important as efficacy and should be carefully analyzed.
Response: Aflibercept has been approved for clinical therapy of wet AMD with minimal inflammation and toxicity issues and investigated for gene therapy as ADMV-022 with excellent safety profile (PMID: 33532145). Our previous studies indicate that anti-Scg3 hFab has a better safety profile than aflibercept (PMID: 35006382). Our unpublished data also revealed that anti-Scg3 hFab inhibits leukocyte and macrophage infiltration and inflammation in the mouse model of laser-induced CNV. Since eye is an immune privileged organ, immune response against AAV vector is unlikely as with other ocular gene therapy studies. Our preliminary data detected no anti-Scg3 hAb antibodies in mouse serum one month post AAV injection. Therefore, our future improvements will focus on AAV2 and higher titer.
- What is the status of injected anti-Scg3 therapies? The field will need to see much more efficacy/safety data from independent labs and in large animal models, probably even in a clinical trial, with a therapeutic protein before considering gene therapy. Any discussion today about anti-Scg3 gene therapies is, in my opinion, premature.
Response: Anti-Scg3 hAb is currently under development toward Investigational New Drug (IND) application and clinical trials for diabetic retinopathy, wet AMD and retinopathy of prematurity. A recent study by an independent group confirmed anti-Scg3 inhibition of corneal neovascularization in rabbits (PMID: 36443679). The therapy is now under investigation in a large animal model by our group.
We agree with the reviewer that anti-Scg3 with no clinical data is far less well-characterized than anti-VEGF. Likewise, protein-based therapies targeting other angiogenic factors, such as endostatin and PEDF, have never been approved or in clinical trial, but gene therapies against these targets are already in Phase 1 or 2 clinical trials to treat wet AMD (PMID: 32649860). Therefore, this first proof-of-concept study only at the preclinical stage should be reasonable.
- The statements regarding the limitations of anti-VEGFs in the introduction and discussion are overly pessimistic compared to reality.
Response: In Introduction, we already recognized that anti-VEGF is superior than thermal laser and photodynamic therapy and that the approval of anti-VEGF is a major breakthrough.
The limited efficacy of anti-VEGF has also been reported in the literature. Otherwise, pharmaceutical industry would not spend big R&D bucks to develop new therapies, such as faricimab, anti-Ang2, anti-PDGF, anti-integrin and anti-S1P (iSONEP), with high attrition rates. However, the underlying mechanisms for poor efficacy to improve visual acuity may not be fully understood. In Discussion, this manuscript speculates the possible adverse effects of anti-VEGF on healthy vessels and neurons that may compromise the efficacy for visual acuity improvement (see our response to Reviewer #1 Comment #1). The discussion is in no way to disparage the importance of anti-VEGF as the current first-line drug therapy for wet AMD therapy.
- Furthermore, it is confusing that the authors suggest an anti-Scg3 + anti-VEGF combination therapy considering the list of concerns about anti-VEGF gene therapies listed in the last paragraph of the discussion.
Response: Combination therapy is well recognized as a valuable strategy to improve efficacy. Anti-VEGF is currently the only approved drug therapy for nAMD. Despite potential safety concerns, combination therapy for anti-Scg3 has no other option but to choose anti-VEGF. However, because of the improvement in treatment efficacy, such combination may reduce dose or treatment frequency that will minimize safety concerns of anti-VEGF.
Additionally, we have discovered second disease-restricted angiogenic factor using our innovative ligandomics technology, as described for Scg3 (PMID: 28330905) and developed neutralizing monoclonal antibody against this novel target. We predict that this new disease-targeted anti-angiogenic therapy will have optimal safety profile similar to anti-Scg3 (PMID: 35006382). We envision that future combination of these two disease-targeted therapies, both selectively inhibiting pathological but not physiological angiogenesis, will synergistically improve treatment efficacy and visual acuity for nAMD patients with minimal adverse effects on healthy vessels and neurons.
Reviewer 3 Report
In this study, Huang et al. developed an AAV-based gene therapy for wet AMD. In a previous study (LeBlanc ME et al. J Exp Med), the same group showed that blocking Scg3 is therapeutic in a mouse of diabetic retinopathy and its action is independent of VEGF pathway. Therefore, anti-Scg3 therapy is a promising alternative for patients with choroideal neovascularisation or diabetic macular edema, and the therapy might have an additive effect.
The current paper aims to establish proof-of-concept for AAV-mediated Scg3 expression. The authors aim to solve to problems with this approach: (1) they claim several patients do not repsond well the anti-VEGF therapy and (2) monthly injections of anti-VEGF is a major treatment burden.
It is important to note that neither anti-Scg3, nor AAV-VEGF is established in the clinic, therfore this approach is unlikely to immeditaely translate to patients. It is unlcear, whether long-term overexpression of Scg3 has a detrimental effect and it would be advisable to develop an antibody agains Scg3 and apply in clinical pratice first. An agent with unknown PD/PK/ADME is unlikely to be applied with an AAV in a first-in-human clinical trial.
However, the experiments clearly show that there is biological effect of an AAV-Scg3 in a mouse model of choroideal neovascularisation.
Comments and questions:
1) It is unclear which cells are secreting the Scg3. The antibody then diffuses to every retinal layers, this has been shown. It has been postulated in other programs that therpeutic antibodies are mostly secreted from the ciliary body. It would be interesting to see mRNA expression (e.g. in situ hybridisation, RNA-Scope) in the eye and identify secreting cells. Also, this assay would allow to detect the decrease of production of anti-Scg3 over time, as described in Figure 4.
2) It is unclear why transgene expression is decreasing over time. What is the explanation for this? Again, expression testing would help to clarify the issue. The immunostaining is not very quantiative.
3) Sometimes the authors say they use the CAG, sometimes they say it’s CBA promtoer. Which one it is?
4) In the introduction it is claimed that only 15-40% of the patients demonstrate increase in BCVA after anti-VEGF therapy in the clinic. This suggests a rather low response rate. In fact, in the majority of patients, there is a response, which is demonstrated by the slower rate of further visual acuity loss, thus the responder rate is higher. Please revise the sentence.
Author Response
Response to Reviewer #3
- It is unclear which cells are secreting the Scg3. The antibody then diffuses to every retinal layers, this has been shown. It has been postulated in other programs that therapeutic antibodies are mostly secreted from the ciliary body.
Response: Endogenous Scg3 is predominantly expressed in neurotransmitter vesicles, including the retinal ganglion cell layer, inner plexiform layer, outer plexiform layer, and photoreceptor inner segments, as previously describe (PMID: PMID: 28330905). In this study, immunohistochemistry mostly detected intracellular transgene proteins, including anti-Scg3 Fab and aflibercept. Because secreted proteins with diffused signals are difficult to be detected due to poor signal contrast. Both transgenes are expressed in almost all retina cells. Secreted transgene proteins from any cells can diffuse extracellularly and regulate endothelial cells to inhibit CNV. We did not analyze transgene expression in the ciliary body. If any, secreted proteins from this tissue will circulate to the anterior humor and trabecular meshwork with minimal therapeutic effect on CNV in the deep retina.
- It is unclear why transgene expression is decreasing over time. What is the explanation for this?
Response: One of the possible explanations is epigenetic regulation at the genome level. We added the following sentence in Discussion. “Reduced transgene expression over time may be due to epigenetic regulations after AAV integration.”
- Sometimes the authors say they use the CAG, sometimes they say it’s CBA promoter. Which one it is?
Response: CAG is used throughout the manuscript.
- In the introduction it is claimed that only 15-40% of the patients demonstrate increase in BCVA after anti-VEGF therapy in the clinic. This suggests a rather low response rate. In fact, in the majority of patients, there is a response, which is demonstrated by the slower rate of further visual acuity loss, thus the responder rate is higher.
Response: These percentages are based on the data of multiple clinical trials in Reference #3. I agree that individual ophthalmologists may have different clinical experience.
Round 2
Reviewer 1 Report
The authors have basically answered the questions. The anti-Scg3 Fab seems like another promising way to inhibit pathological neovascularization.